# Pyrrolizidine Alkaloids Induce Cell Death in Human HepaRG Cells in a Structure-Dependent Manner

**DOI:** 10.3390/ijms22010202

**Published:** 2020-12-28

**Authors:** Josephin Glück, Julia Waizenegger, Albert Braeuning, Stefanie Hessel-Pras

**Affiliations:** 1Department of Food Safety, German Federal Institute for Risk Assessment, Max-Dohrn-Str. 8-10, 10589 Berlin, Germany; josephin.glueck@bfr.bund.de (J.G.); julia.waizenegger@web.de (J.W.); albert.braeuning@bfr.bund.de (A.B.); 2German Nutrition Society, Godesberger Allee 18, 53175 Bonn, Germany

**Keywords:** pyrrolizidine alkaloids, structure dependency, hepatotoxicity, apoptosis

## Abstract

Pyrrolizidine alkaloids (PAs) are a group of secondary metabolites produced in various plant species as a defense mechanism against herbivores. PAs consist of a necine base, which is esterified with one or two necine acids. Humans are exposed to PAs by consumption of contaminated food. PA intoxication in humans causes acute and chronic hepatotoxicity. It is considered that enzymatic PA toxification in hepatocytes is structure-dependent. In this study, we aimed to elucidate the induction of PA-induced cell death associated with apoptosis activation. Therefore, 22 structurally different PAs were analyzed concerning the disturbance of cell viability in the metabolically competent human hepatoma cell line HepaRG. The chosen PAs represent the main necine base structures and the different esterification types. Open-chained and cyclic heliotridine- and retronecine-type diesters induced strong cytotoxic effects, while treatment of HepaRG with monoesters did not affect cell viability. For more detailed investigation of apoptosis induction, comprising caspase activation and gene expression analysis, 14 PA representatives were selected. The proapoptotic effects were in line with the potency observed in cell viability studies. In vitro data point towards a strong structure–activity relationship whose effectiveness needs to be investigated in vivo and can then be the basis for a structure-associated risk assessment.

## 1. Introduction

Pyrrolizidine alkaloids (PAs) are a group of secondary metabolites occurring in a wide variety of plants. These natural toxins are produced and stored in plants to protect them from herbivores. More than 660 PAs and their respective *N*-oxides are known and about half of them exhibit hepatotoxic effects [1]. PAs can be found in more than 6000 plant species, especially in the Boraginaceae, Asteraceae and Fabaceae families, representing approximately 3% of the world’s flowering plants [2,3,4,5].

Due to their widespread distribution and their genotoxic and hepatotoxic properties, PA-containing plants are among the most common poisonous plants that may affect humans, wildlife and livestock [6]. Exposure of humans and livestock to PAs via the consumption of contaminated food and feed may lead to acute and chronic liver damage comprising the induction of hepatic sinusoidal obstruction syndrome, hepatomegaly, ascites and liver hardening, as well as hepatic necrosis, fibrosis and cirrhosis. In 2013, the German Federal Institute for Risk Assessment identified tea and honey as the main sources for human PA exposure in Western countries. Additionally, the consumption of contaminated herbs, flour or grains and herbal medicines can lead to the intake of substantial quantities [1,7,8,9,10,11].

Chemically, PAs consist of a necine base (1-hydroxymethylpyrrolizidine) that can be esterified at the OH-groups at ring positions C-7 and/or C-9 with aliphatic mono- or dicarboxylic acids (necine acids). The group of PAs can be subdivided according to their necine base into the retronecine, heliotridine, otonecine and platynecine types and further by their grade of esterification into free bases, monoesters and cyclic or open-chained diesters. The combination of different necine bases and acids leads to a huge variety of structurally different PAs as summarized in Table 1.

PAs are stored in plants predominantly in their non-toxic *N*-oxidized form but may be reduced during the digestion process by gut bacteria [12,13]. After intestinal uptake, the PA parent compound can be metabolically activated by cytochrome P450 (CYP) enzymes mainly of the 3A and 2B subfamilies in the parenchymal cells of the liver [14]. The generation of highly reactive pyrrole esters leads to the formation of protein and DNA adducts [15,16]. PAs lacking the double bond in the C-1/C-2 position are considered non-hepatotoxic due to the lack of the structural characteristic needed for bioactivation [17,18].

In addition, numerous in vitro studies indicate that monoesters have a significantly lower toxicity than diesters. According to the current state of knowledge, there is still no scientific consensus with regard to the description of structure–effect relationships of PAs. Studies by Merz and Schrenk (2016), who introduced interim relative potency factors based on the latest in vitro and in vivo data, as well as studies from Louisse et al. (2019) and Allemang et al. (2018), who compared cytotoxic and genotoxic effects of several PAs in vitro, constitute a good starting point for further evaluation of relative toxicity potencies [19,20,21]. Nevertheless, there are still many uncertainties about how different PAs act at the molecular level. For example, it is not known whether a grouping of PAs according to their cytotoxic effects can be transferred to other endpoints in the same way, or whether the potencies of different PAs vary between several endpoints. Therefore, we focused on the effects of a set of structurally different PAs on the viability of HepaRG cells and the induction of apoptosis. Based on previous studies on the induction of apoptotic effects with four structurally different PAs, Waizenegger et al. (2018) stated that the toxicity of a PA seems to be dependent on the PA’s necine base. The strongest effects were induced by the cyclic diester senecionine and the open-chained diester echimidine, both belonging to the retronecine-type PAs [22]. To verify this hypothesis, a more extensive set of 22 PAs (Figure 1) was used to investigate the effects in terms of structural properties on the induction of apoptosis in human HepaRG cells.

## 2. Results

### 2.1. PA-Induced Cytotoxic Effects in HepaRG

As a first step, the structure-dependent effects of 22 different PAs on the cell viability of HepaRG cells were analyzed using the MTT cell viability assay (Figure 2). Metabolically competent HepaRG cells were exposed to PAs in concentrations from 0.1 to 250 µM for 24 h. Huge differences in the effects on cell viability between the individual PAs can be seen clearly, as shown on the heat map in Figure 2. The monoesters intermedine, indicine, lycopsamine, rinderine, echinatine and europine, as well as the free bases heliotridine and retronecine, affected the viability of HepaRG cells to no or only to a very little extent (remaining cell viability > 90 %). Open-chained or cyclic diesters induced medium (cell viability between 90 % and 70 %) to strong cytotoxicity (cell viability < 70 %). Platyphylline, considered to be not toxic, showed a weak but statistically not significant decrease in cell viability.

### 2.2. Caspase Induction

The proteolytic cleavage and thereby activation of caspases is a key event in apoptosis induction. Therefore, the effect of PA exposure on the cleavage of the initiator caspases 8 and 9 and the effector caspases 3 and 7 was analyzed via the fluorescence of specific caspase substrates. Since caspase activation is an early event in apoptosis induction, an incubation time of 6 h was chosen according to the previous results of Waizenegger et al. (2018) [22]. In addition, PA concentrations in a range that lead to weak cytotoxic effects of the most cytotoxic PAs (5, 21, 35 µM) were selected to reflect the beginning of apoptosis induction. The initial cell viability screening approach comprised a high number of retronecine-type cyclic diesters and heliotridine-type monoesters. Due to the clear structure-dependent grouping of the PAs in the cell viability assay, the PA test set was now reduced to 14 structurally different PAs representatives, comprising a maximum of three members of the same structural group (underlined in Table 1).

In Figure 3 the cleavage of caspase substrates is depicted as a heat map. In the figure, the PAs are sorted according to their cytotoxic activity (cp. Figure 2). While no caspase activation by the less cytotoxic PAs was detected, a concentration-dependent, up to 5.9-fold increase in caspase 3/7 activity was determined for the more toxic PAs echimidine, seneciphylline, heliosupine, senecionine and lasiocarpine. For caspases 8 and 9, the increase in activity was weaker (up to 1.3- and 2.0-fold) and therefore statistically not significant, even though again the more cytotoxic substances tended to induce a stronger activation of caspases.

### 2.3. Expression Analysis of Apoptosis-Related Genes

Changes in the expression of a set of 16 apoptosis-associated genes were analyzed using quantitative real-time PCR (qPCR). HepaRG cells were treated for 24 h with the test set of 14 PAs in the concentrations also used for analysis of caspase activation (see Section 2.2). The results of gene expression analysis were summarized as a heat map (Figure 4), sorted again by the cytotoxic effects caused by the PAs as shown in Figure 2.

In general, mostly slight alterations were detected at the mRNA level, in line with the fact that apoptosis is predominantly regulated at the protein level by cleavage of target proteins. However, a strong upregulation of the antiapoptotic genes *BCL2* and *BCL2A1* was detected concentration-dependently for the toxic PAs echimidine, seneciphylline, heliosupine, senecionine and lasiocarpine, reaching up to 9.5-fold (*BCL2*, senecionine) and 7.8-fold (*BLC2A1*, senecionine) induction. A slight upregulation of expression was observed for the antiapoptotic gene *BCL2L1* and the proapoptotic genes *BAK1*, *FADD* and *TRAF2*. The gene for apoptotic protease activating factor 1 (*APAF1*) also showed a slight increase in expression after treatment with the more toxic PAs at higher concentrations. For the caspases, with the exception of a slight downregulation of caspase 8 (*CASP8*), no influence on gene expression was detected. The antiapoptotic gene *TRADD*, which is, like *CASP8*, a part of the extrinsic apoptosis pathway, also showed slight downregulation of expression after treatment of HepaRG with some PAs. However, this effect did not seem to be concentration-dependent and not correlated with the toxicity of the corresponding PAs. The same applies to the very slight regulation of *BAD*. No regulation was observed for the antiapoptotic genes *API5*, *BCL2L2* and *MCL1*.

## 3. Discussion

Due to their wide distribution and severe toxicity to wildlife, livestock and humans, PAs have long been known as natural plant toxins. Nevertheless, many details about the toxicological modes of action of PAs are still not fully understood. Moreover, PA content and composition differ between plant species, plant organs, geographical origin and growth state. Environmental factors like climate and soil conditions also have a major impact on PA content and composition [23,24,25]. This additionally aggravates assessment of PA toxicity under real-life conditions.

In risk assessment, it is currently still assumed that all PAs have the same toxicity [26]. However, it is already known that there can be huge differences in toxicity also comprising, for example, phosphorylation of the histone H2AX and micronucleus formation between the different PAs due to structural reasons [19,20,21]. It has been shown that monoesters show hardly any toxic effects, while diesters can cause severe liver damage. Furthermore, 1,2-unsaturated PAs such as platyphylline cannot be metabolically activated and therefore no reactive metabolites are produced that form DNA and protein adducts [17,18]. PA *N*-oxides, which are predominantly found in plants, cannot be directly activated either. However, it is possible that these can be reduced to the PA parent substance during the digestion process [12,13]. Thus, a risk assessment which considers all 1,2-saturated PAs as a homogenous group with equipotent representatives contains a considerable degree of uncertainty, while an approach based on a structure effect-oriented assessment may yield more precise results. In 2016, Merz and Schrenk took a first step towards standardizing the available data and proposed the introduction of relative potency factors for a number of PAs by combining genotoxic, cytotoxic and acute toxic effects in vivo and in vitro [20]. Based on this grouping of PAs according to their toxic potential, a more refined picture of PA toxicity may be developed through targeted studies on various endpoints.

A whole transcriptome analysis by Luckert et al. in 2015 showed dramatic changes in the expression of a wide variety of genes after 24 h of exposure of primary human hepatocytes to the four PAs echimidine, heliotrine, senecionine and senkirkine. Software-based analyses of the regulated genes indicated that, among others, cellular mechanisms such as cell death, apoptosis and necrosis could be impaired [27]. Other studies have also indicated connections between PA exposure and the induction of apoptosis in the liver for the PAs monocrotaline, clivorine, retrorsine and senecionine [28,29,30,31,32]. Based on these data, we focused on structural differences in the induction of cytotoxicity and especially apoptosis by 22 different PAs. Based on previous studies by Waizenegger et al. and Luckert et al. with only four different PAs, we hypothesized that retronecine-type PAs possess the strongest toxic potential [22,27]. This assumption needs to be refined when considering the results of this study: the necine base appears to play a rather minor role in toxicity, as the effects of lasiocarpine and heliosupine (heliotridine type) were largely equivalent to those of senecionine and seneciphylline (retronecine type). Furthermore, we confirmed that PA monoesters and free necine bases induce no to very weak toxicity, while PA diesters have medium to strong toxic potential concerning the induction of cytotoxicity in HepaRG cells.

The grouping of PAs according to their cytotoxic potential was confirmed in the following analysis of caspase activation with a slightly reduced test set. The effector caspases 3 and 7 were substantially activated despite PA concentrations below clearly cytotoxic levels. The activation of the initiator caspases 8 and 9 was less pronounced. PA-induced apoptosis seems to be more likely to occur via the intrinsic signaling pathway (caspase 9-mediated) [28,30,32]. Waizenegger et al. (2018) and Ebmeyer et al. (2019) could nevertheless show that the extrinsic signaling pathway (mediated by Fas via caspase 8) may also be involved. Thus, PA-mediated activation of apoptosis pathways seems to occur via a combination of both extrinsic and intrinsic mechanisms [22,33].

The classification of PAs can also be applied to changes in the expression of apoptosis-associated genes. The effect is particularly evident for the genes encoding the antiapoptotic proteins BCL2 and BCL1A2. The upregulation of the expression of these genes is probably a feedback mechanism of apoptosis induction. The regulation of the apoptotic process mainly takes place at the protein level. The cleavage and degradation of the proteins BCL2 and BCL1A2 decrease the intracellular protein amount and thus their expression is upregulated at the gene level, possibly as part of a feedback mechanism.

Summarizing all the data of this study, the PA with the highest toxic potential are lasiocarpine and heliosupine (both heliotridine-type, open-chained diesters), senecionine and seneciphylline (both retronecine-type, cyclic diesters) and echimidine (retronecine-type, open-chained diesters). Different PA representatives were analyzed in regard to other, more genotoxicity-related endpoints beside apoptosis induction in recent published studies: Louisse et al. detected DNA double-strand break induction by analyzing the phosphorylation of the histone H2AX after exposure of HepaRG cells to 37 different PAs in several concentrations [19]. The micronucleus assay as an endpoint for genotoxicity was performed by Allemang et al. for 15 different PAs and PA *N*-oxides in HepaRG cells [21]. Rutz et al. (2020) also analyzed 11 PAs with regard to the induction of cytotoxicity and the formation of micronuclei in CYP3A4-overexpressing HepG2 cells, as well as the induction of mutations using the Ames fluctuation test [34]. Gao et al. investigated the cytotoxicity of 10 PAs in primary rat hepatocytes [35]. Lester et al. investigated the formation of DHP–DNA adducts in rat hepatocytes after exposure to nine different PAs [36]. In all the studies mentioned, heliosupine, lasiocarpine, senecionine and seneciphylline, if used, showed very strong toxic effects compared to the effects of other PAs, which fits to the results of this study. The strength of toxicity of echimidine varies between the publications but is nevertheless always in the medium toxicity range. Minor differences between the effects of PAs can most likely be attributed to different test systems (cell lines), solvents or tested endpoints. The quantity and activity of xenobiotic-metabolizing enzymes with their metabolic activation and detoxification capacities, for example, have a direct influence on the toxicity of PAs [37]. This is why differences between the individual cell types and culture conditions are particularly important here. However, despite the differences in the test set-up, only minor deviations in the effects of most PAs are shown between the different studies. Since the effects are largely comparable, the sorting of PAs according to their cytotoxic potential can be considered as reliable for in vitro analyses with cytotoxic or genotoxic endpoints.

Although the results of the studies are broadly similar for many PAs, there are large differences for some PAs such as monocrotaline. For this PA, a clear discrepancy can be seen between the relative potency factors according to Merz and Schrenk [20] and the above-mentioned in vitro studies. The introduction of relative potency factors is therefore controversial and must be considered critically. There is already a large amount of toxicity data on various PAs from studies that have been conducted in vivo or in vitro. Unfortunately, these data can often not be compared directly with each other because different PAs were used in different species and/or cell lines, with different incubation times, doses or concentrations, and different endpoints or methods. In addition, the comparison between in vitro and in vivo data is difficult because aspects such as metabolism and uptake in the digestive tract, distribution in the body and excretion are not considered in in vitro studies. Furthermore, metabolic activation in the liver and other organs cannot be mapped 1:1 in vitro either. Even very metabolically competent cell lines often show a deficiency of certain enzymes [38,39,40].

Previous investigations on the structure dependence of PA toxicity in vitro are not sufficient for use in risk assessment. For a reliable assessment of the risk of PA contamination in food to humans, some gaps need to be closed. In order to refine the classification of PAs according to their toxic potential, it is necessary to examine structure-dependent toxicokinetic properties and toxification reactions in more detail. It is therefore necessary to extend in vitro and in vivo studies to enable a precise derivation of potency factors to refine the assessment of risk of PA exposure to humans.

## 4. Materials and Methods

### 4.1. Chemicals

All PAs used in this study were purchased from Phytoplan Diehm & Neuberger GmbH (Heidelberg, Germany), except platyphylline which was obtained from BOC Sciences (New York, NY, USA). The purity of all PAs was at least 95%. PAs were stored at −20 °C dissolved in 50% Acetonitrile (ACN)/50% H_2_O as 5 mM stock solutions. All other chemicals were obtained from Merck (Darmstadt, Germany) or Sigma-Aldrich (Taufkirchen, Germany) in the highest purity available.

### 4.2. HepaRG Cell Culture

The human hepatoma cell line HepaRG was obtained from Biopredic International (Saint-Gregoire, France). After seeding, the cells were cultivated for two weeks in William’s Medium E with stable glutamine (PAN Biotech, Aidenbach, Germany) supplemented with 10% fetal bovine serum (FBS Good forte, PAN-Biotech, Aidenbach, Germany), 100 U/mL penicillin and 100 µg/mL streptomycin (PAN-Biotech, Aidenbach, Germany), 5 µg/mL human insulin (PAN-Biotech, Aidenbach, Germany) and 50 µM hydrocortisone hemisuccinate (Sigma-Aldrich, Taufkirchen, Germany). After this proliferation phase, differentiation was initiated by adding 1.7% dimethyl sufoxide (DMSO). HepaRG cells were fully differentiated after two weeks. All experiments with HepaRG were performed with differentiated cells seeded at passages 16 to 20.

### 4.3. Cell Viability Assay

HepaRG cells were seeded at a density of 9000 cells per well in the inner 60 wells of a 96-well plate. After four weeks of cultivation, DMSO and FBS in the cell culture medium were set to 0.5% and 2% for 48 h before incubation, respectively, to optimize the effects on gene expression. The cells were treated with different PA concentrations for 24 h as indicated in the figures according to Waizenegger et al. (2018) [22]. Due to the low solubility of some PA, concentrations higher than 250 µM would lead to cytotoxic concentrations of the solvent ACN. For the cell viability assay, 10 µL undiluted MTT (3-(4,5-dimethylthiazol-2-yl)-2,5-diphenyltetrazolium bromide) reagent per well was added. After 30 min of incubation at 37 °C, the supernatant was discarded and the violet formazan crystals were dissolved in 130 µL isopropanol with 0.7% sodium dodecyl sulfate (SDS) per well (30 min, shaking protected from light). Absorption was determined at 570 and 630 nm (reference wavelength) [41].

### 4.4. Total RNA Isolation and Quantitative Real-Time Polymerase Chain Reaction (qPCR)

To investigate effects of PA treatment on the expression of apoptosis-associated genes, gene expression was analyzed using quantitative reverse transcription real-time PCR. HepaRG cells were seeded in 6-well plates at a density of 0.2 × 10^6^ cells per well. DMSO and FBS were set to 0.5% and 2% for 48 h before PA treatment to optimize inducibility of gene expression. The cells were incubated with different PA concentrations for 24 h. Afterwards, cells were washed twice with cold PBS. The RNeasy Mini Kit (Qiagen, Hilden, Germany) was used for total RNA extraction following the manufacturer’s instructions including the on-column DNA digestion step. RNA concentration and purity were measured at 260, 280 and 230 nm on a TecanM200Pro spectrophotometer using a NanoQuant plate. The ratios A260/A280 and A260/A230 were calculated to detect potential contaminants.

For cDNA synthesis, the High Capacity cDNA Reverse Transcriptase Kit (Applied Biosystems, Foster City, CA, USA) was used as recommended by the manufacturer with 1 µg RNA per reaction. For qPCR analysis 1 µL cDNA, 5 µL Maxima SYBR Green/ROX qPCR Master Mix (Thermo Fisher Scientific, Waltham, MA, USA) and 300 nM of each primer were used in a total volume of 10 µL per reaction. In each PCR run, at least three wells without a template were used as control. The following thermal profile was used: initial denaturation (15 min, 95 °C), 40 cycles of denaturation (30 s, 95 °C) and annealing/elongation (1 min, 60 °C), final elongation (10 min, 60 °C). The specific amplification of a single PCR product was confirmed by the analysis of the recorded dissociation curve. The cDNA amplification of three replicates per sample was detected at the 7900HT Fast Real-Time PCR System (Applied Biosystems, Foster City, CA, USA) in 384-well format. The Ct values were normalized to the housekeeping gene *GUSB* (β-glucuronidase), evaluated according the 2^−ΔΔCt^ method [42] and referred to solvent control. The primer sequences are summarized in the Appendix A. *GUSB* was used as a housekeeping gene with a constitutive expression after validation for the HepaRG cell line and the exposure to PAs.

### 4.5. Activation of Caspases

The caspase activation assay was performed as described previously [22]. Briefly, differentiated HepaRG cells were incubated with PAs for 6 h in 96-well plates. After treatment, cells were lysed using 50 mM HEPES buffer (pH 7.4) containing 2% Triton X-100. Specific fluorogenic substates for caspases 3/7 (Ac-DEVD-AFC), caspase 8 (Ac-IETD-AFC) and caspase 9 (Ac-LEHD-AFC) were added to the cells for 1 h (37 °C). The fluorescence of the cleaved substrates was detected at λ_ex_ = 380 and λ_em_ = 500 nm (DEVD and IETD), or at λ_ex_ = 400 and λ_em_ = 505 nm (LEHD).

### 4.6. Statistical Analysis

Statistical analysis was performed using Sigmaplot 14.0 software (Systat Software, Erkrath, Germany). To determine statistical differences in a concentration series, one-way analysis of variance (ANOVA) followed by the Dunnett’s post hoc test versus the respective solvent control was performed. Statistically significant differences are indicated by * *p* < 0.05, ** *p* < 0.005, *** *p* < 0.001.

## Figures and Tables

**Figure 1 ijms-22-00202-f001:**
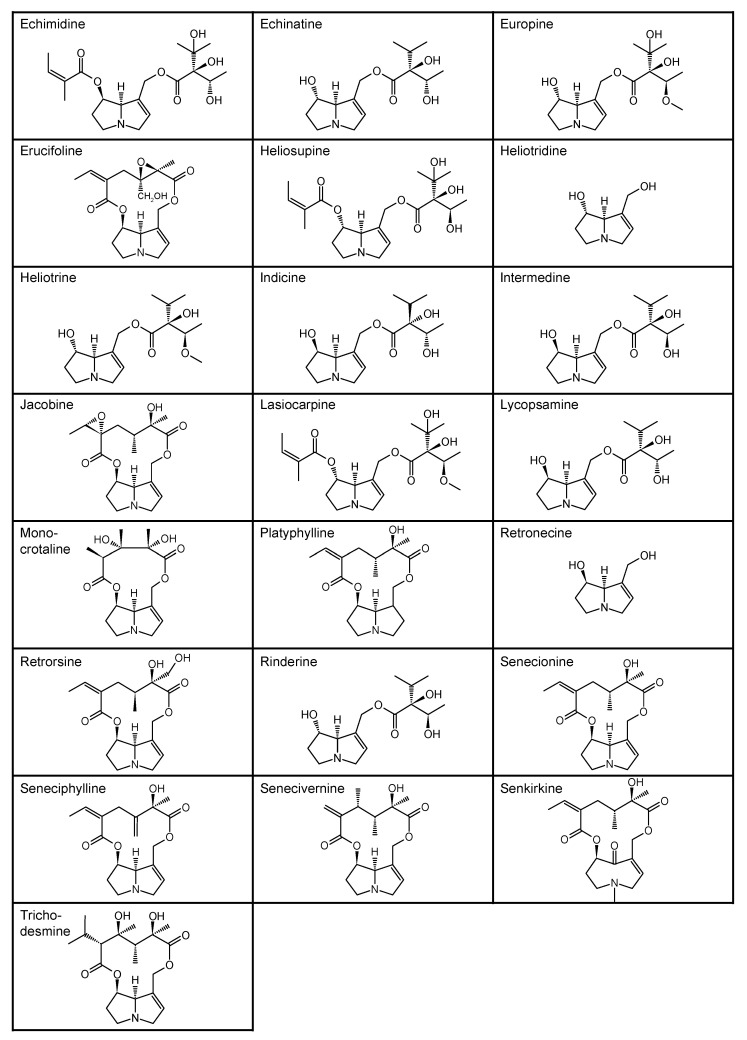
Structures of the 22 pyrrolizidine alkaloids used in this study.

**Figure 2 ijms-22-00202-f002:**
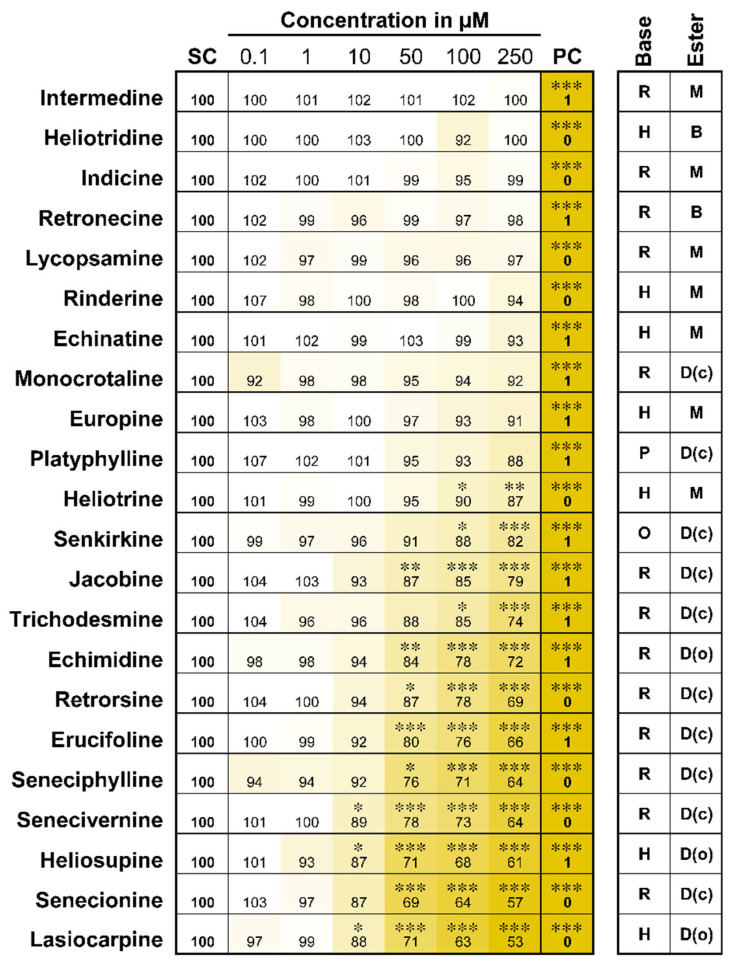
Decrease in viability of HepaRG cells after 24 h of PA treatment. Cell viability was measured by the MTT assay. Triton X-100 (0.05 %) was used as positive control (PC); the solvent control (SC) contained 2.5 % ACN and 0.5 % DMSO. Data are shown as means of at least three independent replicates normalized to the solvent control. Numbers show the cell viability in percent of the solvent control. The stronger the decrease of cell viability the darker is the yellow coloration in the figure. Values and standard deviations can be found in the Appendix A. Statistics: * *p* < 0.05, ** *p* < 0.005, *** *p* < 0.001 (one-way ANOVA followed by Dunnett‘s post hoc test versus the solvent control). Abbreviations for structural characteristics: retronecine (R), heliotrine (H), otonecine (O) or platynecine (P) type; free base (B), monoester (M), open-chained diester (D(o)) or cyclic diester (D(c)).

**Figure 3 ijms-22-00202-f003:**
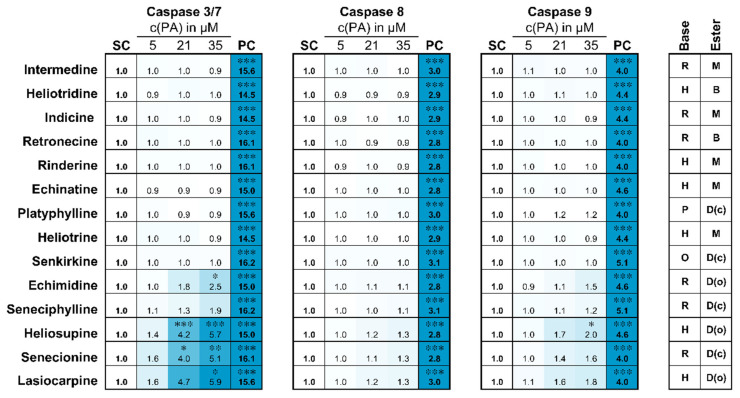
Activation of caspases 3/7, 8 and 9 in HepaRG cells after PA treatment for 6 h. Differentiated HepaRG cells were treated with 5, 21 and 35 µM of each PA for 6 h. Caspase activities were measured in cell lysates by detecting the cleavage of the specific fluorogenic substrates Ac-DEVD-AFC (caspase 3/7), Ac-IETD-AFC (caspase 8) and Ac-LEHD-AFC (caspase 9). Staurosporine (5 µM) was used as positive control (PC) for apoptosis induction; the solvent control (SC) contained 0.35 % ACN and 0.5 % DMSO. Data are shown as means of three independent replicates. Values indicate the x-fold activation of the caspases normalized to solvent control. The stronger the caspase activity the darker is the blue coloration in the figure. White fields mean no activation of caspases. Mean values and standard deviations can be found in the Appendix A. Statistical analysis: * *p* < 0.05, ** *p* < 0.005, *** *p* < 0.001 (one-way ANOVA followed by Dunnett‘s post hoc analysis versus the solvent control). Abbreviations for structural characteristics: retronecine (R), heliotrine (H), otonecine (O) or platynecine (P) type; free base (B), monoester (M), open-chained diester (D(o)) or cyclic diester (D(c)).

**Figure 4 ijms-22-00202-f004:**
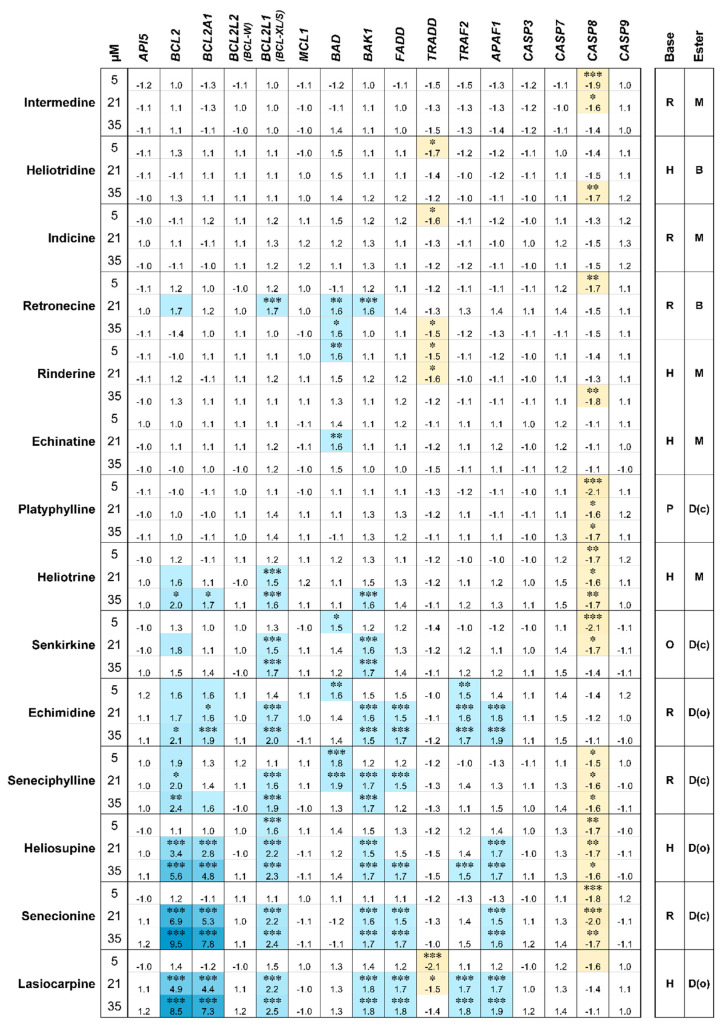
Changes in gene expression levels of apoptosis-associated genes after 24 h PA exposure of differentiated HepaRG. The results were evaluated according the 2^−ΔΔCt^ method, normalized to the gene expression of the housekeeping gene *GUSB* (β-glucuronidase) and referred to untreated cells (solvent control, 0.35% ACN, 0.5% DMSO). Values are shown as fold change compared to the solvent control and as means of three replicates. The cut-off for gene regulation was set from −1.5 to 1.5. Changes in regulations between −1.5- and 1.5-fold to SC were considered not to be biologically significant. The higher the downregulation of gene expression the darker the yellow, the higher the upregulation the darker the blue coloration. White fields in the figure indicate no gene regulation in the defined fold change cut-off.Mean values and standard deviations are summarized in the Appendix A. Statistics: * *p* < 0.05, ** *p* < 0.005, *** *p* < 0.001 (one-way ANOVA followed by Dunnett‘s post hoc analysis versus the respective solvent control). Structural characteristics: retronecine (R), heliotrine (H), otonecine (O) or platynecine (P) type; free base (B), monoester (M), open-chained diester (D(o)) or cyclic diester (D(c)).

**Table 1 ijms-22-00202-t001:** Overview of different pyrrolizidine alkaloids (PAs) and their structural characteristics. For cytotoxicity studies, all listed esters and the free bases heliotridine and retronecine were used. The underlined PAs represent the reduced test set for all following experiments.

Necine Base		Monoesters	Open-Chained Diesters	Cyclic Diesters
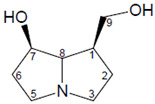	Platynecine (7R) 1,2-saturated			Platyphylline
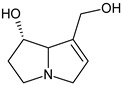	Heliotridine (7S) 1,2-unsaturated	Echinatine Europine Heliotrine Rinderine	Heliosupine Lasiocarpine	
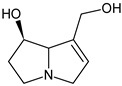	Retronecine (7R) 1,2-unsaturated	Indicine Intermedine Lycopsamine	Echimidine	Erucifoline Jacobine Monocrotaline Retrorsine Senecionine Seneciphylline Senecivernine Trichodesmine
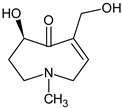	Otonecine (7R) 1,2 unsaturated			Senkirkine

## Data Availability

The data presented in this study are available in the Appendix A which can be found at www.mdpi.com/xxx/s1.

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
