# Peer review of "Pyrrolizidine Alkaloids Induce Cell Death in Human HepaRG Cells in a Structure-Dependent Manner"

_ijms, 2020, doi:10.3390/ijms22010202_

Round 1

Reviewer 1 Report

Reviewing for the International Journal of Molecular Sciences, ijms-1035856: «Pyrrolizidine alkaloids induce cell death in human HepaRG cells in a structure-dependent manner” from J. Glück et alii.

In this manuscript, J. Glück and coworkers explored the pro-apoptotic activity of 22 different mono- and di-esters of pyrrolizidine alkaloids (PA). PA are an important family of vegetal compounds present in several thousand plants. Esters of PA are considered as the most toxic in the family. Their activity was tested on Hepa-RG cells, a well differentiated cell line still expressing various cytochromes. Authors tested the cytotoxicity by MTT test on 22 PA, while activation of 3 types of caspases by fluorogenic tests and expression of 16 genes controlling apoptosis was explored on a subset of 14 PA.

The paper is clear, well written, experiments and conclusions are sound.

The authors focused on apoptosis only without considering the possible intervention of others cell death mechanisms such as autophagy, intrinsic necroptosis, anoïkis, etc…Why?

The presentation of gene expression changes is difficult to understand on figure 4 that is not a real heatmap but rather a table in disguise. Where are the fold changes? Please correct.

Author Response

Response to reviewer 1 comments´:

Comment 1:

The authors focused on apoptosis only without considering the possible intervention of others cell death mechanisms such as autophagy, intrinsic necroptosis, anoïkis, etc…Why?

Response:

Thank you very much for your constructive comment. We decided to focus on apoptosis due to previous results generated in our group. We already saw evidence for the induction of apoptosis in the transcriptomic study of Luckert et al. (2015) performed in primary human hepatocytes with 4 model PA. In the following studies by Waizenegger et al. (2018) and Ebmeyer et al. (2019) the endpoint apoptosis was further investigated but only with a limited number of PA. Therefore, in this study, we now aimed to investigate these observations further in terms of structural dependence. Thus, we focused only on the induction of apoptosis and not on other cell death mechanisms. Additionally, in the previous studies, there was no evidence (esp in the transcriptomic approach by Luckert et al.) that other cell death mechanisms like autophagy, intrinsic necroptosis, and anoikis may be affected.

Comment 2:

The presentation of gene expression changes is difficult to understand on figure 4 that is not a real heatmap but rather a table in disguise. Where are the fold changes? Please correct.

Thank you very much for this comment. We corrected the Figure 4 as recommended. We added the fold changes to the heatmap.

Reviewer 2 Report

GENERAL COMMENTS:

The work presented in this paper assessed the cytotoxicity of pyrrolizidine alkaloids in a hepatic cell line to associated level of toxicity with structure of the plant secondary metabolites. I thought that the work was well done and the manuscript well written. I had only a few minor questions and comments that I would like the authors to consider prior to publication.

I really liked the way that the authors presented the data in this manuscript. I thought that the heat maps were a great way to not only consolidate the work of the 22 PA evaluated, but to provide a visual element to the empirical data as well.

Because this study has sought to determine the impact of pyrrolizidine alkaloids on cell death, is apoptosis or programmed cell death the correct term for this mechanism being studied? Is necrosis more appropriate, since induction of cell death is caused by a toxin in the work presented? I realize that some of the assays were apoptosis-specific, but how do you know which mechanism was involved in the cell viability assay?

SPECIFIC COMMENTS:

L81 – How were these concentration ranges initially selected for the MTT assay?

L110 - How were these concentration ranges initially selected for the capsase activation assay?

L170 – Huge differences in what … response or potency? The reader is left to guess the authors intentions as written here.

L294-304 – Was RNA quality assessed? It is fairly common practice to validate the use of a single housekeeping gene. How was this reference gene selected and was it validated in this cell line previously? Were there no-template controls included in the analysis of gene expression? What about melt curve analysis to confirm that only a single product produced in the reaction well? I guess with these questions, I am suggesting that a little more detail be provided in this section.

Reviewer 3 Report

In the present manuscript, Hessel-Pras and her co-workers have shown in a very careful study that pyrrolizidine alkaloids can induce cell death of human HepaRG cells and that this cell death is dependent on the structure of the pyrrolizidine derivatives. Especially open-chained and cyclic heliotridine and retronecine diesters caused severe cytotoxic effects finally leading to cell death, whereas the action of the monoesters on HepaRG cells did not influence their viability. This outcome was further confirmed by caspase activation and gene expression analysis and the observed proapoptotic effects were in agreement with the cell viability investigation. The conclusion of their study is that there is a strong structure–activity relationship for the toxicity of pyrrolizidine alkaloids which should be studied in vivo. The manuscript has been written very well and the text is virtually free of mistakes. However, in the introduction to the field (first two paragraphs on page 1), I have noted that a recent and comprehensive review on pyrrolizidine alkaloids which covered the occurrence, biosynthesis, toxicity, biological activity and chemistry of pyrrolizidine alkaloids surprisingly has not been cited. This article should be added to the reference list: Tamariz, J.; Burgueno-Tapia, E.; Vázquez, M.A.; Delgado, F. Pyrrolizidine alkaloids. In The Alkaloids – Chemistry and Biology, Vol. 80, Knölker, H.-J., Ed.; Academic Press: London, 2018; pp. 1-331. With this minor revision, this excellent manuscript is strongly recommended for publication.

Author Response

Response to reviewer 3 comments´:

Comment 1:

However, in the introduction to the field (first two paragraphs on page 1), I have noted that a recent and comprehensive review on pyrrolizidine alkaloids which covered the occurrence, biosynthesis, toxicity, biological activity and chemistry of pyrrolizidine alkaloids surprisingly has not been cited. This article should be added to the reference list: Tamariz, J.; Burgueno-Tapia, E.; Vázquez, M.A.; Delgado, F. Pyrrolizidine alkaloids. In The Alkaloids – Chemistry and Biology, Vol. 80, Knölker, H.-J., Ed.; Academic Press: London, 2018; pp. 1-331. With this minor revision, this excellent manuscript is strongly recommended for publication.

Response:

Thank you for your comments. We added the reference in the introduction (end of first paragraph, line 41, ref. 11) as recommended by the reviewer.